# STELAR: Dual-space training EEG Foundation Models for Transferable Representations

## Abstract

Electroencephalography (EEG) is a non-invasive technique that provides critical insights for diagnosing neurological disorders. However, leveraging EEG in machine learning remains challenging due to its inherently low signal-to-noise ratio (SNR), pronounced inter-subject variability, and heterogeneous channel configurations across datasets. These issues make it challenging to design a general-purpose encoder that can reliably capture robust and transferable EEG representations. Most existing EEG foundation models adopt self-supervised learning frameworks, typically pairing a primary encoder with several auxiliary components. While these auxiliary modules are intended to support representation learning, in practice, they often dominate the optimization process, preventing the encoder from developing strong, generalizable features. Consequently, even well-trained models may fall short in downstream applications. To address this limitation, we propose STELAR, a novel EEG foundation model that concentrates training on the encoder while minimizing the role of auxiliary components. STELAR introduces a three-part dual-space pretraining strategy that integrates representation-space alignment with lightweight signal-space reconstruction: (i) visible-token alignment directly supervises encoder outputs, (ii) masked-token alignment enforces generative consistency through a compact prediction head, and (iii) linear masked reconstruction preserves fidelity to the original signals. This streamlined design substantially reduces auxiliary parameters while yielding a cleaner and more effective pretraining pipeline compared to prior approaches. In addition, STELAR incorporates a spatio-temporal cross-attention encoder, which jointly captures spatial dependencies across EEG channels and temporal dynamics across time. Empirical results demonstrate that STELAR converges rapidly, within 15 epochs, and consistently outperforms previous EEG foundation models by up to 5% under linear probing evaluation. All source code will be released publicly upon acceptance of this work.

## 1 Introduction

Understanding the brain has long been a central scientific pursuit. Brain–computer interfaces (BCIs) leverage electroencephalography (EEG) to non-invasively record neural activity, yet EEG signals remain difficult to interpret due to non-stationarity and low signal-to-noise ratio. Prior work has developed task-specific methods for motor imagery (Altaheri et al., 2021; Dai et al., 2020), emotion recognition (Dadebayev et al., 2022; Gao et al., 2024), seizure detection (Ahmad et al., 2022; Yıldız et al., 2022), and sleep staging (Phan & Mikkelsen, 2022; Wang et al., 2024b; Zhou et al., 2024), showing that EEG carries rich cognitive and clinical information. Yet, these methods remain fragmented, each tied to a single task. This raises a key question: *How can we build a unified model that learns generalizable EEG representations transferable across tasks?* To this end, we introduce STELAR (**S**patio-**T**emporal **E**ncoder-centric **L**ightweight **A**lignment & **R**econstruction), a framework that advances EEG foundation models and establishes a unified evaluation protocol, addressing a critical gap in prior work.

Developing universal EEG foundation models is challenging due to several factors: the inherently low signal-to-noise ratio of EEG, high variability across and within subjects (Saha & Baumert, 2020; Del Pup et al., 2025; Rezzouk et al., 2025), and differences in electrode montages across datasets (Han et al., 2025). In addition, many recent approaches follow monolithic designs where large auxiliary modules (e.g., decoder, predictor) dominate learning (Jiang et al., 2024; Wang et al., 2024a), making

it difficult to ensure that the encoder itself captures generalizable representations. These challenges have motivated a range of self-supervised strategies proposed in the literature, which we review in Section 2. Building on these insights, we introduce STELAR, a streamlined framework that addresses these issues by centering representation learning in the encoder while keeping auxiliary components lightweight.

In this work, we present STELAR, an encoder-centric pre-training framework that seamlessly unifies representation-space alignment with lightweight signal-space reconstruction. Building on the strengths of recent EEG foundation models, STELAR enhances efficiency and stability through a three-part dual-space objective: (i) **visible-token alignment**, which directly supervises encoder outputs, (ii) **masked-token alignment**, which promotes generative consistency via a compact predictor, and (iii) **linear masked reconstruction**, which preserves fidelity to raw signals. Integrated within a Criss-Cross attention backbone (Wang et al., 2024c), these components together yield an encoder that is both computationally efficient and capable of producing high-quality, transferable representations.

The experimental analysis validates our design: STELAR converges within a few epochs during pre-training (Appendix B), reduces pre-training parameters by nearly 50% compared to EEGPT-like framework (Section 3.4), and achieves stronger linear probing performance than prior foundation models across benchmarks (Table 7). Our ablation results highlight the necessity of visible-token alignment and lightweight reconstruction for achieving stable representation learning (Section 4.4).

By shifting the heavy lifting of representation learning to the encoder and standardizing the evaluation protocol, STELAR establishes a principled path for future EEG foundation models. Its lightweight and stable design lowers the computational barrier to entry, enabling broader adoption in neuroscience and BCI research. More importantly, strong linear probing performance across benchmarks demonstrates that STELAR learns representations that are not only good in performance but also robust and generalizable, paving the way toward reliable foundation models for clinical and cognitive EEG applications.

**Our contributions can be summarized as follows:**
- **Encoder-centric pre-training.** STELAR emphasizes token-level supervision with minimal auxiliary modules, ensuring that representation quality resides in the encoder.
- **Three-part dual-space objective.** Visible-token alignment, masked-token alignment, and linear masked reconstruction act together to encourage stability, fidelity, and transferability.
- **Efficiency with strong transfer and generalizability.** STELAR significantly reduces auxiliary parameters for pretraining, converges in a few epochs, and consistently achieves strong linear probing results across multiple EEG benchmarks.

## 2 RELATED WORK

**Generative self-supervised learning.** BENDR (Kostas et al., 2021) followed the masked autoencoder (MAE) scheme, which randomly masked a proportion of the input and trained the model to reconstruct this information. However, BENDR did not account for the correlation of different channels of the EEG signal. To address this limitation, LaBram (Jiang et al., 2024) introduced a learned neural tokenizer to discretize the EEG signal into discrete tokens for MAE learning. CbraMod (Wang et al., 2024c) proposed a Criss-Cross attention to aggregate information within the same channel and across different channels of multi-channel EEG signals. Although these methods achieve good performance when fine-tuned on downstream tasks, training with MAE can heavily emphasize on low-level signal details, which might not be essential for effective representation learning (Grill et al., 2020). This phenomenon is also observed in EEG signals, as evidenced by substantial declines in linear probing performance (Jiang et al., 2024; Wang et al., 2024c; Xiong et al., 2025).

**Discriminative self-supervised learning.** Among discriminative methods, contrastive learning (CL) is currently widely used (Mohsenvand et al., 2020; Yang et al., 2023). In CL, an encoder is trained to maximize the similarity between "positive pairs", and minimize the similarity of the "negative pairs". (Mohsenvand et al., 2020) trained a channel feature extractor by extending the SimCLR framework to EEG data. (Yang et al., 2023) addressed the mismatched channel between different samples by masking some channels, and applied contrastive learning to learn the core representation of multichannel EEG signals. However, EEG signals are noisy, vary across and within subjects, and lack clear unit boundaries, making it challenging to definitively determine which pairs are "positive"

or "negative". This is problematic as CL relies on well-defined positive and negative pairs to learn meaningful representations (Kalantidis et al., 2020; Lan et al., 2023).

**Hybrid self-supervised learning.** (Lee et al., 2024; Zhu et al., 2023) proposed to apply both masked reconstruction loss and contrastive loss to utilize the advantage of both types of loss. So far, (Grill et al., 2020; Caron et al., 2021) have shown the potential of alignment loss, which uses only "positive pairs", eliminating the need for proper "negative pairs" in CL. In particular, alignment loss aligns the online encoder with the exponential moving average (EMA) target encoder, under the hypothesis that the EMA representation is more stable and prevents model collapse. Motivated by this idea, EEGPT (Wang et al., 2024a) proposed a dual loss scheme, combining alignment loss and masked reconstruction loss, for EEG signal. With this dual-loss scheme, EEGPT achieved better performance compared to other EEG foundation models in the linear probing setup (Wang et al., 2024a; Xiong et al., 2025).

Our method is inspired by EEGPT but extends it significantly with several key enhancements. First, we introduce an additional alignment-based loss to strengthen representation learning. Second, we tailor an improved architecture that more effectively captures the unique characteristics of EEG signals. Finally, we propose a novel encoder-centric framework, minimizing the auxiliary modules, that ensures representation quality resides in the encoder.

# 3 METHOD

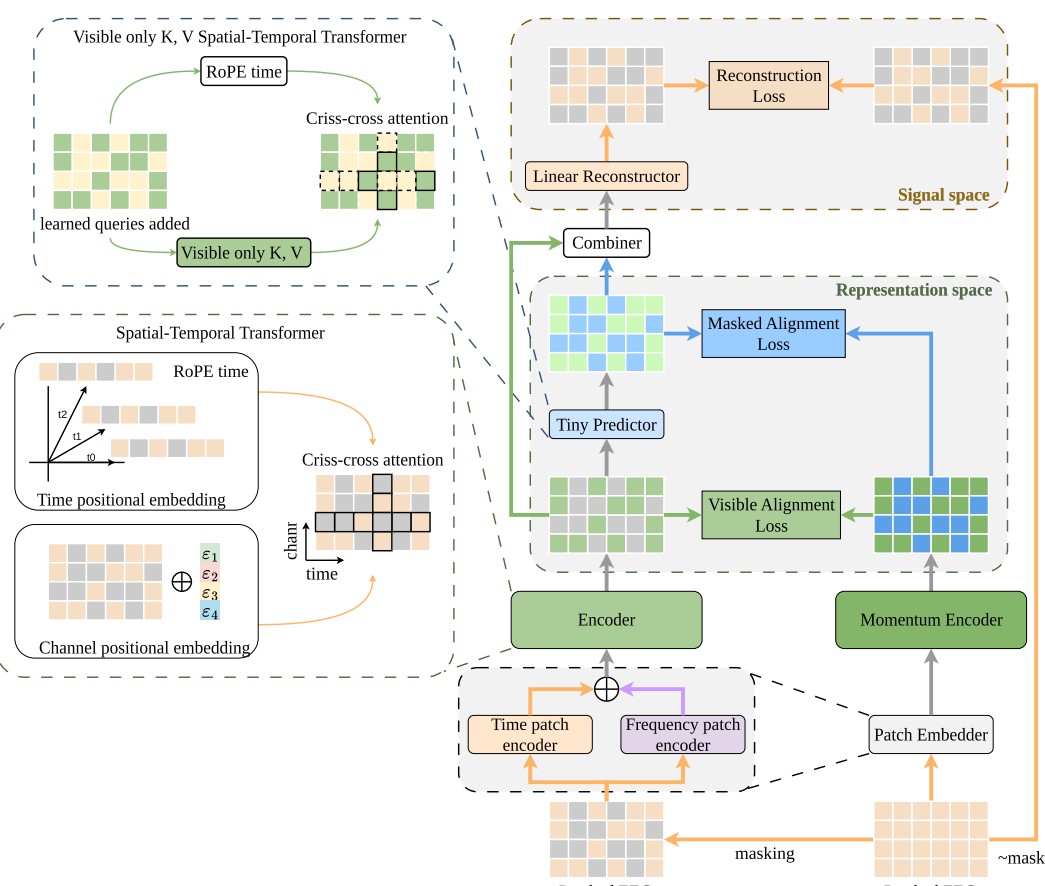

Figure 1: Schematic of the STELAR framework, including the patch embedder with Criss-Cross attention; *two* representation-space alignments (visible pre-predictor; masked post-predictor); *lightweight* signal-space reconstruction; and EMA branch.

The STELAR pre-training framework (Figure 1) consists of an encoder, momentum encoder, predictor, and reconstructor. EEG signals are patchified (30s windows) with a high masking ratio, and patch embeddings are learned via transformer blocks with Criss-Cross attention. Training follows a dual-space self-supervised scheme: visible representation alignment, masked representation alignment with a lightweight predictor, and mask-based reconstruction with a linear layer. The resulting model is evaluated by linear probing across diverse downstream tasks.

### 3.1 PATCH ENCODING

#### 3.1.1 PATCHING & MASKING

Consider a multichannel EEG input $S \in \mathbb{R}^{C \times T}$, where $C$ is the number of electrodes and $T$ the number of time points. We divide $S$ into temporal windows of length $t$, forming patches $X \in \mathbb{R}^{C \times n \times t}$ with $n = \lfloor T/t \rfloor$ per channel. Each patch $x_{i,j} \in \mathbb{R}^t$ yields the sequence $X = \{x_{i,j}\}_{i=1..C, j=1..n}$ of length $Cn$. To enable training with partial information, we apply a random mask $\mathcal{M} = \{m_{i,j}\}$, with $m_{i,j} \sim \text{Bernoulli}(r)$. If $m_{i,j} = 0$, the original patch is kept; otherwise it is replaced by a constant mask token $x_M \in \mathbb{R}^t$:

$$\tilde{x}_{i,j} = \begin{cases} x_{i,j}, & m_{i,j} = 0, \\ x_M, & m_{i,j} = 1. \end{cases} \quad (1)$$

The masked sequence is then $\tilde{X} = \{\tilde{x}_{i,j}\} \in \mathbb{R}^{C \times n \times t}$.

#### 3.1.2 TIME–FREQUENCY PATCH EMBEDDING

EEG signals exhibit both temporal dynamics and oscillatory patterns across frequency bands. To exploit this complementary structure, we follow CBraMod (Wang et al., 2024c) and encode each masked patch $\tilde{x}_{i,j}$ with a dual-branch encoder. **Time branch** includes 1D convolutional blocks (conv–norm–GELU) produce a temporal embedding $e_{i,j}^t \in \mathbb{R}^{d_e}$. **Frequency branch** includes fast Fourier transform (FFT) followed by a fully connected layer yields a spectral embedding $e_{i,j}^f \in \mathbb{R}^{d_e}$. The final representation is the sum:

$$e_{i,j} = e_{i,j}^t + e_{i,j}^f, \quad E = \{e_{i,j}\} \in \mathbb{R}^{C \times n \times d_e}. \quad (2)$$

#### 3.1.3 CHANNEL EMBEDDING

EEG signals are montage-dependent, so preserving channel identity is essential. Following EEGPT (Wang et al., 2024a), we assign each channel $c_i$ a trainable embedding $\epsilon_i \in \mathbb{R}^{d_e}$, added to its patch embeddings $E$:

$$E^o = E + \{\epsilon_i\}_{i=1}^C \in \mathbb{R}^{C \times n \times d_e}. \quad (3)$$

This montage-aware design introduces a lightweight spatial bias and improves transferability: non-standard channel sets in downstream datasets can be aligned via a simple adapter (e.g., $1 \times 1$ projection) without retraining.

### 3.2 REPRESENTATION-SPACE ALIGNMENT

#### 3.2.1 SPATIO–TEMPORAL ENCODER

Our encoder follows CBraMod (Wang et al., 2024c), which uses *Criss-Cross attention* to jointly model temporal and spatial dependencies across patches. This design better captures intra-channel dynamics and cross-channel interactions than standard attention, aligning with Diver-0 (Han et al., 2025), which also stresses the importance of explicit cross-channel modeling for generalization. We therefore adopt criss-cross attention, as neighboring temporal and spatial patches share stronger relationships than distant ones.

Different from the Absolute Channel Positional Encoding (ACPE) (Wang et al., 2024c), which fixes the receptive field, we combine (i) learnable channel embeddings for spatial identity and (ii) RoPE (Su

et al., 2024) for temporal order. This allows flexible dependency modeling across arbitrary spans and channel subsets (within the 10–20 system), while keeping the pre-training pipeline clean and lightweight.

**Transformer Block.** Each block consists of layer normalization, Criss-Cross attention, a residual connection, and a feed-forward layer. Given patch embeddings $E^o \in \mathbb{R}^{C \times n \times d_e}$, we first normalize to obtain $\tilde{E}$, then enrich with RoPE and channel embeddings:

$$\hat{E} = \tilde{E} + \text{RoPE}_{\text{time}}(\tilde{E}) + \text{ChanEmbed}(\tilde{E}). \tag{4}$$

**Criss-Cross attention.** The attention module consists of two branches: *spatial attention* (across channels at the same time step) and *temporal attention* (across time within the same channel). The enriched input $\hat{E}$ is projected into $K$ heads, with half assigned to spatial and half to temporal attention:

$$\text{head}_k = \begin{cases} \text{S-Attention}_k(\hat{E}), & k \le K/2, \\ \text{T-Attention}_k(\hat{E}), & k > K/2. \end{cases} \tag{5}$$

The outputs are concatenated as

$$\text{Criss-cross-Attention}(\hat{E}) = \text{Concat}(\text{head}_1, \ldots, \text{head}_K). \tag{6}$$

**Output Representation.** Stacking such blocks yields

$$E^r = \{e^r_{i,j} \mid i \in [1, \ldots, C], \, j \in [1, \ldots, n]\} \in \mathbb{R}^{C \times n \times d}, \tag{7}$$

where each token $e^r_{i,j}$ integrates patch content, temporal context, and channel identity under the Criss-Cross attention structure.

### 3.2.2 TINY PREDICTOR.

Let $E^r = \{e^r_{i,j}\} \in \mathbb{R}^{C \times n \times d}$ be encoder outputs, with $\mathcal{V}$ and $\mathcal{M}$ denoting visible and masked positions. Masked tokens are reconstructed using a tiny predictor $\text{Pred}$ (1–2 transformer layers), ensuring representational capacity resides in the encoder, consistent with findings from BYOL (Grill et al., 2020). Each masked position $(i, j) \in \mathcal{M}$ is assigned a query

$$q_{i,j} = q_\star + \epsilon_i + \text{RoPE}_{\text{time}}(j), \tag{8}$$

where $q_\star$ is a learnable mask embedding, $\epsilon_i$ the channel embedding, and RoPE encodes temporal order. Queries attend only to visible tokens $E^r_{\mathcal{V}}$ (*visible-only K/V attention* (He et al., 2022; Fu et al., 2024)), improving stability and efficiency.

The predictor thus outputs

$$\{\text{pred}_{i,j}\}_{(i,j) \in \mathcal{M}} = \text{Pred}(\{q_{i,j}\}_{(i,j) \in \mathcal{M}}, E^r_{\mathcal{V}}). \tag{9}$$

By keeping $\text{Pred}$ shallow, the encoder is forced to embed sufficient information in visible tokens, while queries gain minimal inductive cues from RoPE and channel embeddings.

### 3.2.3 MOMENTUM ENCODER.

We use a momentum encoder $f_\xi$, identical to the online encoder $f_\theta$, updated via EMA:

$$Z^{\text{enc}} = f_\theta(\tilde{X}), \quad Z^{\text{mom}} = f_\xi(X), \quad \xi \leftarrow \tau\xi + (1-\tau)\theta, \tag{10}$$

where $X$ is the original input and $\tilde{X}$ its masked version (with constant, non-learnable tokens). This avoids information leakage while keeping input structure consistent.

The momentum encoder serves as a slowly updated teacher, providing stable asymmetric targets, akin to BYOL (Grill et al., 2020) and MoCo (He et al., 2020). Building on these methods, our framework introduces *visible pre-predictor alignment*, which is described next.

### 3.2.4 REPRESENTATION-SPACE DUAL ALIGNMENT.

We define two alignment objectives against the momentum encoder.

**Visible alignment.** For visible tokens $(i, j) \in \mathcal{V}$, encoder outputs are directly matched to momentum features:

$$\mathcal{L}_{\text{vis}} = \frac{1}{|\mathcal{V}|} \sum_{(i,j) \in \mathcal{V}} \|\text{LN}(e_{i,j}^r) - \text{stopgrad}(\text{LN}(z_{i,j}^{\text{mom}}))\|_2^2, \tag{11}$$

ensuring the encoder, not the predictor, captures discriminative features.

**Mask alignment.** For masked tokens $(i, j) \in \mathcal{M}$, predictor outputs are aligned with momentum features:

$$\mathcal{L}_{\text{mask}} = \frac{1}{|\mathcal{M}|} \sum_{(i,j) \in \mathcal{M}} \|\text{LN}(\text{pred}_{i,j}) - \text{stopgrad}(\text{LN}(z_{i,j}^{\text{mom}}))\|_2^2. \tag{12}$$

Here the predictor is purely generative, reconstructing from visible context. This dual scheme differs from prior BYOL/MoCo setups by directly supervising visible tokens. The representation loss is

$$\mathcal{L}_{\text{rep}} = \lambda_v \mathcal{L}_{\text{vis}} + \lambda_m \mathcal{L}_{\text{mask}}. \tag{13}$$

## 3.3 SIGNAL-SPACE MASKED RECONSTRUCTION

To complement representation-space alignment, we reconstruct only masked patches. Predicted and visible features are merged via a combiner, then projected back:

$$\hat{x}_{i,j} = W_{rec} \, \text{Combiner}(\text{pred}_{i,j}, \text{enc}_{\text{vis}}), \quad (i, j) \in \mathcal{M}, \tag{14}$$

with loss

$$\mathcal{L}_{\text{rec}} = \frac{1}{|\mathcal{M}|} \sum_{(i,j) \in \mathcal{M}} \|\hat{x}_{i,j} - x_{i,j}\|_2^2. \tag{15}$$

The full objective is

$$\mathcal{L} = \lambda_v \mathcal{L}_{\text{vis}} + \lambda_m \mathcal{L}_{\text{mask}} + \lambda_r \mathcal{L}_{\text{rec}}, \tag{16}$$

balancing alignment and reconstruction while keeping the predictor shallow and the encoder central.

## 3.4 COMPLEXITY

A key principle of our framework is a *clean, lightweight pre-training pipeline*. By avoiding heavy decoders or large predictors, training converges in few epochs, requires fewer resources, and reduces cost and environmental impact. Formally, the parameter count is

$$\text{Params}_{\text{STELAR}} \approx 2P_{\text{enc}} + P_{\text{pred}}^{\text{tiny}} + P_{\text{rec}}^{\text{lin}} \approx 2P_{\text{enc}}, \tag{17}$$

as the tiny predictor and linear reconstructor are negligible relative to the encoder.

This yields fewer parameters and faster training than frameworks with heavy decoders/predictors (Wang et al., 2024a) or separately trained tokenizers (Jiang et al., 2024).

## 3.5 LINEAR-PROBING METHOD

For downstream evaluation, we freeze the pretrained encoder and train only a lightweight head, ensuring that performance reflects the quality of the learned representations. We exploit the full patch embeddings $E^r \in \mathbb{R}^{C \times n \times d}$, enabling the probe to aggregate features (via pooling or projections) or derive task-specific summary tokens, thereby capturing both global and fine-grained dynamics. The probing network applies lightweight spatial filters ($1 \times 1$ convolutions) for channel alignment, and a linear classifier. As in EEGPT (Wang et al., 2024a), we utilize a channel adapter to map heterogeneous electrode sets into the pre-trained montage, thereby enabling broad dataset compatibility. This design maintains the probe's lightweight nature while faithfully assessing encoder generality and transferability.

## 4 EXPERIMENTS

### 4.1 PRETRAINING DATA & SETUP

**Pre-training Dataset.** STELAR is pre-trained on a curated subset of TUEG (Obeid & Picone, 2016), excluding TUAB (Lopez de Diego et al., 2017) and TUEV (Golmohammadi et al., 2017) to avoid leakage. After extensive preprocessing to remove noise and artifacts, the dataset is reduced to $\sim$500 hours of clean EEG (see Appendix D).

**Preprocessing & Settings.** Following prior work (Wang et al., 2024c), we harmonize sampling rates, select 19 standard channels, apply filtering, window EEG into 30s segments, and normalize to $[-1, 1]$. Pre-training uses random masking (70%), AdamW optimization, and 5 model variants (Table 5). More implementation details and scaling results are in Appendix D.

### 4.2 DOWNSTREAM, EVALUATION & SETUP

**Evaluation Protocol.** To ensure fair evaluation, we adopt subject-wise cross-validation, reporting averages across folds. Different from many prior works that report validation results only, our scheme isolates test sets entirely (see Appendix E).

**Downstream Tasks.** We evaluate on **5 tasks** across **6 datasets** (Table 7), covering motor imagery, seizure detection, sleep staging, abnormality detection, and error-related potentials. All data are resampled to 200Hz, normalized to $[-1, 1]$, and mapped to the predefined channel set. Dataset-specific preprocessing and truncation are described in Appendix E.

**Baselines.** We compare STELAR to state-of-the-art EEG foundation models (EEGPT, LaBraM, CBraMod), using their official setups and preprocessing pipelines for fair comparison (see Appendix E).

**Downstream Setup.** We use linear probing (frozen encoder + linear head). For multiclass tasks, we report Balanced Accuracy, Cohen's Kappa, and Weighted F1; for binary tasks, Balanced Accuracy, AUC-PR, and AUROC. STELAR employs a lightweight channel adapter as in EEGPT. Full setup details are in Appendix E.

### 4.3 RESULTS

Table 1 shows that STELAR matches or outperforms prior EEG foundation models across six benchmarks, with clear gains on BCIC-2A, PhysioNet-MI, Sleep-EDFx, and TUAB, and competitive results on KaggleERN and TUEV. It performs especially well on long-range data (Sleep-EDFx, 30 s/sample) while remaining strong on short, event-based tasks, indicating robust modeling of both long- and short-term EEG structure. These results are obtained without any pretraining on the downstream datasets, highlighting the effectiveness of the architecture. Notably, the tiny variant often rivals or exceeds the base model, suggesting compact encoder-centric designs are sufficient, with further gains likely from broader pretraining.

### 4.4 ABLATIONS

**Ablation study on pre-training**

Table 2 shows that pre-training clearly enhances downstream performance: balanced accuracy improves by over 3% absolute, with consistent gains in Cohens's kappa and F1. This confirms that encoder representations learned through our dual-space pretraining transfer effectively, even in the tiny-scale model. The improvement highlights the value of representation-space alignment and signal-level reconstruction in capturing generalizable EEG features.

**Ablation study on different pre-training methods**

Table 3 shows that all three loss components contribute to final performance. Dropping either $\mathcal{L}_{\text{vis}}$ or $\mathcal{L}_{\text{mask}}$ reduces balanced accuracy by 1–2%, showing that visible and masked alignment are

---

[1]Note that the EEGPT pre-training corpus includes **PhysioNet-MI**, while CBraMod was pre-trained on data including **TUAB** and **TUEV**, which overlap with our downstream evaluation.

Table 1: Linear probing results across four downstream tasks[1].

| Datasets | Methods | Balanced Accuracy | Cohen's Kappa/ AUC-PR | Weighted F1/ AUROC |
|---|---|---|---|---|
| BCIC-2A | LaBraM | 0.4667 | 0.2889 | 0.4476 |
| | EEGPT | 0.5495 | 0.3993 | 0.5322 |
| | CBraMod | 0.5133 | 0.3511 | 0.5011 |
| | **Ours-tiny** | 0.5799 | 0.4399 | **0.5690** |
| | **Ours-base** | **0.5803** | **0.4404** | 0.5684 |
| PhysioNet-MI | LaBraM | 0.2766 | 0.0356 | 0.2626 |
| | EEGPT | 0.5319 | 0.3758 | 0.5360 |
| | CBraMod | 0.4625 | 0.2835 | 0.4561 |
| | **Ours-tiny** | 0.5603 | 0.4138 | 0.5611 |
| | **Ours-base** | **0.5688** | **0.4295** | **0.5766** |
| KaggleERN | LaBraM | 0.4997 | 0.7090 | 0.4954 |
| | EEGPT | **0.5632** | 0.7938 | **0.6397** |
| | CBraMod | 0.5018 | 0.7320 | 0.5298 |
| | **Ours-tiny** | 0.5414 | **0.7994** | 0.6273 |
| | **Ours-base** | 0.5520 | 0.7980 | 0.6325 |
| Sleep-EDFx | LaBraM | 0.5721 | 0.5360 | 0.6402 |
| | EEGPT | 0.6131 | 0.6149 | 0.6953 |
| | CBraMod | 0.6420 | 0.6280 | 0.7041 |
| | **Ours-tiny** | 0.6889 | **0.6623** | **0.7480** |
| | **Ours-base** | **0.6970** | 0.6590 | 0.7455 |
| TUEV | LaBraM | 0.4372 | 0.5025 | 0.7366 |
| | EEGPT | 0.5173 | 0.5101 | 0.7480 |
| | CBraMod | 0.3796 | 0.4734 | 0.7162 |
| | **Ours-tiny** | **0.5361** | 0.5704 | 0.7876 |
| | **Ours-base** | 0.5014 | **0.5720** | **0.7909** |
| TUAB | LaBraM | 0.7315 | 0.7958 | 0.7989 |
| | EEGPT | 0.7762 | 0.8593 | 0.8561 |
| | CBraMod | 0.5914 | 0.5685 | 0.6230 |
| | **Ours-tiny** | **0.8009** | 0.8511 | **0.8752** |
| | **Ours-base** | 0.7987 | **0.8630** | 0.8731 |

Table 2: Ablation on pre-training with STELAR-tiny.

| Method | | BICIC-2A | | | TUEV | | |
|---|---|---|---|---|---|---|---|
| | | Bal. Acc.(%) | Kappa(%) | F1(%) | Bal. Acc.(%) | Kappa(%) | F1(%) |
| STELAR-w/o pre-training | | 0.5449 | 0.3932 | 0.5346 | 0.4973 | 0.5148 | 0.7590 |
| **STELAR** | | **0.5799** | **0.4399** | **0.5690** | **0.5361** | **0.5704** | **0.7876** |

Table 3: Ablation study on pre-training losses with STELAR-tiny

| Method | BCIC-2A | | |
|---|---|---|---|
| | Bal. Acc.(%) | Kappa(%) | F1(%) |
| STELAR-w/o $\mathcal{L}_{vis}$ | 0.5637 | 0.4183 | 0.5546 |
| STELAR-w/o $\mathcal{L}_{mask}$ | 0.5597 | 0.4130 | 0.5448 |
| STELAR-w/o $\mathcal{L}_{rec}$ | 0.5401 | 0.3868 | 0.5259 |
| **STELAR** | **0.5799** | **0.4399** | **0.5690** |

both necessary for robust representation learning. Removing $\mathcal{L}_{rec}$ yields the most significant drop, confirming that a lightweight reconstruction term is critical for preserving fidelity of EEG dynamics.

The full objective, combining all terms, achieves the strongest downstream accuracy, validating our encoder-centric dual alignment with reconstruction design.

## 4.5 EFFICIENCY, STABILITY & SCALING

STELAR is designed to be highly parameter- and compute-efficient. Compared with existing EEG foundation models, our encoder-centric design with lightweight auxiliary modules achieves superior performance while requiring two orders of magnitude fewer parameters and FLOPs (see Appendix B).

Scaling analysis further reveals that accuracy improves from tiny to little, but saturates beyond the base variant. Larger models obtain lower pre-training loss but transfer poorly, indicating overfitting to limited EEG data and a mismatch between model capacity and dataset scale (Kaplan et al., 2020; Zhai et al., 2022; Hoffmann et al., 2022). Compact variants (tiny–base) thus strike the best trade-off between efficiency and robustness (see Appendix B).

Finally, the pre-training curves (Figure 2, Appendix) confirm the stability of our pipeline: convergence occurs within 15 epochs, showing that dual-space alignment with lightweight reconstruction provides a clean learning signal and enables efficient optimization without long training schedules. This demonstrates that streamlined pre-training not only reduces computation and energy costs but also yields high-quality EEG representations.

## 4.6 LIMITATION

While STELAR demonstrates strong efficiency and generalization, several limitations remain. First, the scale of pre-training data is modest compared to the vision or language domains, which limits scalability; larger and more diverse EEG corpora are needed to fully realize the potential of foundation models. Second, STELAR still requires a channel adapter to handle varying electrode montages across datasets, which introduces an additional adaptation step and prevents strict permutation-equivariance. Third, although our design achieves stable convergence, it still depends on a momentum encoder and predictor, which adds complexity to the pipeline.

## 5 CONCLUSION

We presented STELAR, an encoder-centric EEG foundation model that integrates dual-space alignment with lightweight reconstruction. Our encoder-centric design emphasizes token-level supervision while keeping auxiliary modules minimal, enabling fast convergence, reduced complexity, and strong transfer. Linear probing evaluation conducted across diverse BCI tasks demonstrates that even the lightweight variants can achieve good performance, robust generalizability, and transferability. STELAR thus establishes a principled step toward robust and accessible EEG foundation models, paving the way for broader adoption in neuroscience and clinical applications. In future work, we aim to extend this framework to larger and more diverse pre-training corpora and explore multimodal integration.

## REPRODUCIBILITY STATEMENT

We have made every effort to ensure the reproducibility of our work. Details on data preprocessing pipelines, pretraining configurations, and downstream evaluation setups are provided in Appendix D and Appendix E respectively. These resources together enable others to replicate our experiments and verify the reported findings.

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

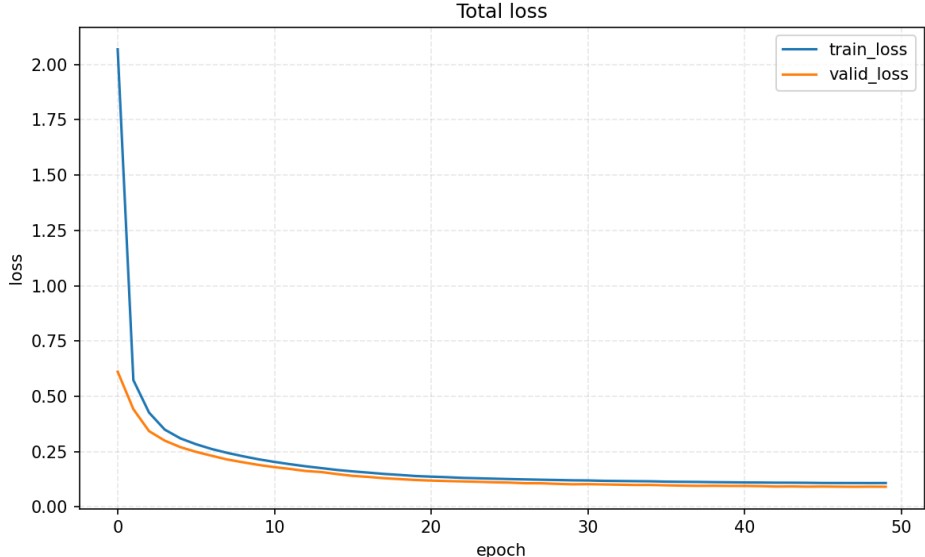

Figure 2: Pretraining loss curve of the STELAR-huge model over 50 epochs. The model converges around epoch 15, demonstrating efficient and stable representation learning.

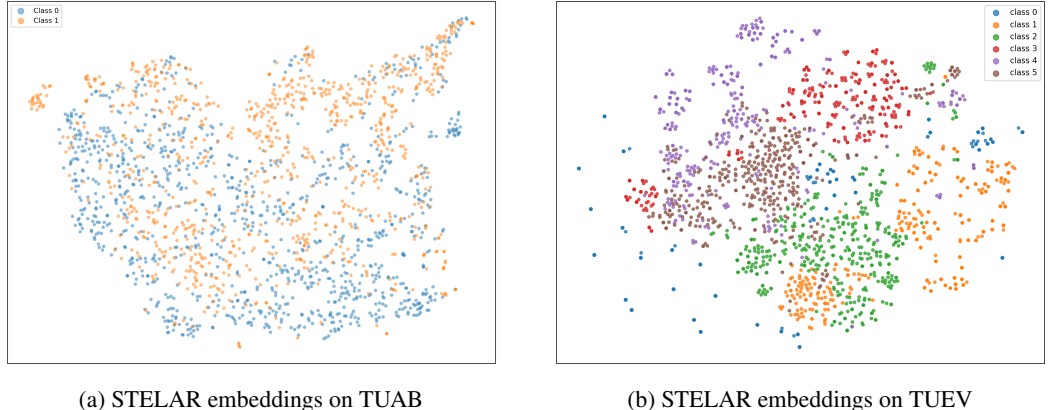

(a) STELAR embeddings on TUAB          (b) STELAR embeddings on TUEV

Figure 3: t-SNE visualizations of STELAR embeddings on (a) TUAB and (b) TUEV.

# A    VISUALIZATION

The pre-training loss curve of our STELAR-huge model is shown in Figure 2. Over the 50-epoch pre-training process, the loss exhibits a consistent downward trend. Convergence is reached around epoch 15, after which the loss stabilizes at a low level. This behavior indicates that the model rapidly learns meaningful representations from EEG data, and the training process remains stable throughout the full pre-training trajectory.

We provide a visualization of the STELAR encoder representations. As shown in Figure 3a and Figure 3b, STELAR successfully clusters the representations of the TUAB and TUEV datasets. Although there are some areas where the embeddings are not fully clustered, STELAR demonstrates a strong ability to generate representations that can effectively classify between different classes.

Table 4: Pretraining computation cost

| Method | Total Params | Encoder Params | MFLOPs/step | BCIC-2A-BAC |
|---|---|---|---|---|
| LaBraM | 5.992M | 5.8M | 2,583 | 0.4667 |
| EEGPT | 76M | 25M | 25,000 | 0.5495 |
| CBraMod | 3.95M | 3.9M | 6,940 | 0.5133 |
| **Ours-tiny** | 161K | 124K | 619 | 0.5799 |
| **Ours-base** | 6.72M | 5.4M | 7,511 | 0.5803 |

Table 5: STELAR model variants and their parameter sizes.

| Variants | Embed dim | Depth | Num heads | Encoder params | Pretrain loss | BCIC-2A-BAC |
|---|---|---|---|---|---|---|
| tiny | 64 | 2 | 2 | 124K | 0.2725 | 0.5799 |
| little | 128 | 8 | 2 | 1.4M | 0.1672 | 0.5827 |
| base | 256 | 8 | 4 | 5.4M | 0.1297 | 0.5803 |
| large | 512 | 8 | 8 | 21.3M | 0.1013 | 0.5652 |
| huge | 512 | 12 | 8 | 31.8M | 0.0911 | 0.5652 |

# B  EFFICIENCY, STABILITY & SCALING

Table 4 shows that STELAR is highly efficient: the tiny variant uses two orders of magnitude fewer parameters and FLOPs than EEGPT yet achieves the best balanced accuracy on BCIC-2A. This confirms that an encoder-centric design with lightweight auxiliary modules yields superior representations without the overhead of heavy decoders or predictors, an important advantage for EEG research where resources are limited.

Scaling analysis in Table 5 reveals that accuracy improves from tiny to little, but saturates beyond the base variant. Larger variants achieve lower pre-training loss but overfit and transfer poorly, reflecting a mismatch between model capacity and the limited size/noisiness of current EEG corpora (Kaplan et al., 2020; Zhai et al., 2022; Hoffmann et al., 2022). Compact variants (tiny, base) therefore offer the best trade-off between capacity and robustness, showing that bigger is not necessarily better for EEG foundation models.

The learning curves in Figure 2 in the Appendix confirm the stability of our pre-training pipeline. The rapid convergence within 15 epochs shows that the dual-space alignment with lightweight reconstruction provides a clean learning signal, enabling efficient optimization without the need of long schedules. This further supports our claim that a streamlined pre-training design can save both computing time and energy while still producing high-quality EEG representations.

# C  ADDITIONAL RESULTS

## C.1  ABLATION STUDY ON MASK RATIO

In this section, we investigate the impact of different mask ratios during pretraining. We pretrain STELAR with various mask ratios and then evaluate its performance on the BCIC-2A dataset in the downstream task. The balanced accuracy varies slightly between different masked ratios. With higher mask ratios during pretraining, the encoder requires less computational cost to process the visible patches. Therefore, we select a 70% mask ratio to balance between training performance and computational efficiency.

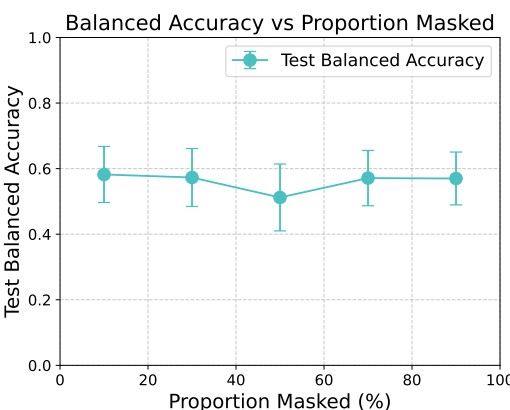

Figure 4: STELAR's performance on BCIC-2A based on masked ratio during pretraining

## D  PRETRAINING DATA & SETUP

**Pre-training Dataset.** STELAR is pre-trained on a subset of TUEG (Obeid & Picone, 2016), which is the largest publicly available dataset, with a total duration of 27,062 hours. This data set was recorded on the standard 10-20 system. Most of the recordings are made at 256 Hz. To avoid information leaks during downstreaming on TUAB (Lopez de Diego et al., 2017) and TUEV (Golmohammadi et al., 2017), the STELAR pre-training data set excludes these two corpora from TUEG. Hence, our raw pre-training dataset has 1,840 hours of EEG recordings. The TUEG dataset is heavily affected by data contamination, with large amounts of unannotated noise, artifacts, and malfunctioning channels. Hence, STELAR employed an efficient preprocessing strategy to eliminate noise in the pre-training data.

**Preprocessing.** Our preprocessing strategy is similar to CBraMod (Wang et al., 2024c) for minimizing variability caused by inconsistent recording and eliminating data noise. First, the recording scheme of TUEG may be prone to contamination, especially at the beginning and end of the recording. Therefore, the one-minute beginnings and endings of all recordings are discarded to remove low-quality data. All recordings were recorded with a 10-20 standard system with 40 common channels. Second, to remove noise from degraded channels without missing important data sources, we only selected 19 most common channels (Fp1, Fp2, F7, F3, Fz, F4, F8, T3, C3, Cz, C4, T4, T5, P3, Pz, P4, T6, O1, O2). Third, because EEG signals focus on a small range of frequencies, EEG recordings were then filtered with a band-pass filter (0.3Hz - 75Hz) to remove noise, along with a notch filter at 60Hz to discard the power line noise. Long-duration samples potentially capture the long-term dependency of EEG signals. Therefore, all recordings were then resampled at 200Hz and cut into 30-second windows without overlap to create uniformly long-term samples. Raw EEG is often disturbed by non-brain physiological activities such as ECG, EOG, and EMG (Yin et al., 2025), which can potentially create "excessive pulses" that deceive models as abnormal recordings. For robust and stable pre-training, we finally eliminated samples with a maximum voltage amplitude exceeding $100\mu V$, before normalizing all samples to a range of -1 to 1. After this preprocessing phase, our pre-training dataset was reduced by 3.5 times from 1,840 hours to 500 hours. STELAR demonstrates efficiency despite its small-scale pre-training.

**Pre-training Settings.** STELAR was implemented with PyTorch 2.1.2 and CUDA 11.8. After preprocessing, each 30s EEG segment, sampled at 200 Hz with 19 channels, is regarded as one sample. We define the length of each patch as 200 time points, equivalent to 1s EEG data. Patches were randomly masked with a ratio of 70%. 10% of the pretraining dataset was held out for validation. Pretrained model was optimized by AdamW optimizer with learning rate $1e-4$ for 50 training epochs using a batch size of 128. We provide 5 architectural variants outlined in Table 5), all trained on NVIDIA A100 40GB GPUs. **The pre-training time of STELAR-huge variant is 4.2 hrs with $2\times$ A100 40GB GPUs**.

We next describe the experimental settings used for STELAR pre-training. In line with CBraMod (Wang et al., 2024c), all EEG recordings are segmented into 30-second samples. This

Table 6: Hyperparameters for STELAR pre-training.

| Component | Hyperparameters | Settings |
|---|---|---|
| EEG sample | Channels | 19 |
| | Time points | 6000 (30s @ 200Hz) |
| | Patch dimension | 200 |
| | Sequence length | 30 patches |
| | Mask ratio | 0.7 |
| | Mask token | Constant |
| Encoder | Layers (depth) | 2–12 (variant dependent) |
| | Hidden dimension | 64–512 |
| | Attention heads | 2–16 |
| | MLP ratio | 4.0 |
| | $qkv$ bias | True |
| | Init std | 0.02 |
| | Norm $\epsilon$ | $1 \times 10^{-6}$ |
| | Attention pattern | Criss-Cross |
| Predictor | Depth | 1–2 layers |
| | KV mode | Global visible-only K/V |
| | Attention | Criss-Cross |
| Reconstructor | Head | Linear (hidden dimension, patch dimension) |
| Momentum Encoder | EMA momentum | Cosine ramp $0.996 \rightarrow 0.9995$ |
| Pre-training | Epochs | 50 |
| | Batch size | 64 (global) |
| | Optimizer | AdamW |
| | Learning rate | $1 \times 10^{-4}$ |
| | LR schedule | Warmup + Cosine decay |
| | Warmup steps | $\max(500, 0.1T)$ |
| | Weight decay | $5 \times 10^{-2}$ (cosine schedule) |
| | Adam $\beta$ | (0.9, 0.999) |
| | Adam $\epsilon$ | $1 \times 10^{-8}$ |
| | Gradient clipping | 1.0 |
| | Loss weights | $\lambda_{\text{rec}} = 10.0, \lambda_{\text{vis}} = 1.0, \lambda_{\text{mask}} = 1.0$ |

window length is notably longer than that employed in prior work, such as BIOT (Yang et al., 2023) (10 seconds) and LaBraM (Jiang et al., 2024) (4–8 seconds). We choose 30-second segments for two main reasons: (1) they provide the model with longer temporal contexts, enabling the capture of long-term dependencies that have been shown to improve downstream performance (Kostas et al., 2021); and (2) the 30-second duration closely matches the segment lengths commonly used in the downstream tasks evaluated in this study, ensuring consistency between pre-training and fine-tuning. A complete list of pre-training hyperparameters is provided in Table 6.

# E DOWNSTREAM, EVALUATION & SETUP

**Comprehensive Evaluation.** We observe that the performance of EEG models is strongly affected by subject partition during evaluation, which leads to considerable variability across different selections (Del Pup et al., 2025; Rezzouk et al., 2025). However, many existing EEG models employ a mixed split scheme, often involving samples from the test set into the training or validation sets, which may prevent them from revealing their true potential and pose a challenge in model comparison. For fair evaluation, we have extensively built a **subject-wise cross-evaluation** scheme, in which all subjects are partitioned into $N$ folds for the validation set or the test set. For example, we conduct $N$ fine-tunings; in each of them, one fold is held out as the test set while the remaining folds are used for training and validation. After each fine-tuning time, we just once test the "best validated checkpoint", which is defined by the training checkpoint with the highest monitoring metric on the validation set

Table 7: Overview of EEG datasets and their corresponding BCI tasks.

| BCI Task | Dataset | Rate | #Channels | Duration | Labels |
|---|---|---|---|---|---|
| Motor Imagery | BCIC-2A | 250 Hz | 22 | 4s | 4-class |
| | PhysioNet-MI | 160 Hz | 64 | 4s | 4-class |
| Event-related Potentials | KaggleERN | 200 Hz | 56 | 2s | 2-class |
| Sleep Staging | Sleep-EDFx | 100 Hz | 2 | 30s | 5-class |
| Seizure / Event Detection | TUEV | 250 Hz | 16 | 10s | 4-class |
| Abnormal EEG Detection | TUAB | 250 Hz | 16 | 10s | 2-class |

(Cohen's kappa for multiclassification and AUC-PR for binary classification). Reported performance is the average across all folds. Details about train-validation-test splits of each dataset are presented below.

**Downstream BCI Tasks & Preprocessing.** To comprehensively evaluate STELAR, we conducted experiments on **5 tasks** using **6 different datasets**, as presented in Table 7. These datasets were recorded at various sampling rates with varying numbers of channels. To efficiently adapt pre-trained STELAR and create a universal downstreaming framework, we resample all samples to 200 Hz and scale them to the range $[-1, 1]$ as done with the pre-training dataset. Due to mis-match channel numbers between the pretraining dataset and various downstream datasets, we constructed a linear mapping to map the dataset's channels to the pre-defined channels, similar to EEGPT. Each dataset associated with a downstreaming task has specific events with different time spans so we adaptively truncate them accordingly to capture meaningful EEG samples. More details about downstreaming datasets are also presented in Table 7

**Baselines.** Existing state-of-the-art EEG foundation models, such as EEGPT, LaBraM, and CBraMod, are regarded as our baselines. We used their best performance reported in the original works for comparison. Firstly, we preprocess the datasets following the papers' specifications, i.e., applying their corresponding sampling rates, channels, band-pass filters, sample lengths, etc. Secondly, we apply their corresponding setups for each downstream dataset, including learning rate, masking ratio, and specific additional architectures (for example, EEGPT requires an additional adapter before the encoder).

**Downstream Setup.** We conducted linear probing evaluation scheme (freezing encoder + fine-tuning linear head) on the six downstream datasets. Balanced Accuracy, Cohen's Kappa, and Weighted F1 are reported for multiclass classification (BCIC2A, PhysioNet-MI, Sleep-EDFx, and TUEV), while Balanced Accuracy, AUC-PR, and AUROC are reported for binary classification (TUAB, KaggleERN). STELAR uses a linear channel adapter to adapt the specific channels of the downstream dataset to predefined channels, similar to EEGPT. For the baselines, the setups from their original works are used.

**Common settings.** All experiments are performed with a global batch size of 64 with seed set to 7. Optimization uses AdamW with a maximum learning rate of $5e-4$ and weight decay of 0.05, following a OneCycle schedule with 20% warm-up.

### E.1 BCIC-2A

**Description & Preprocessing.** BCIC-2A consists of data from 9 subjects doing trials of 4 different motor imagery tasks. These tasks are motor imagery of the left hand (Class 1), right hand (Class 2), feet (Class 3), and tongue (Class 4). Each subject performs two sessions on different days, with each session consisting of 288 trials. STELAR applies a band-pass filter from 0 to 38 Hz, sampling rate at 200 Hz, and 4-second window sample (800 data points).

**Evaluation.** We adopt a leave-one-subject-out (LOSO) cross-validation protocol. We perform 9 fine-tunings, each involving a different subject as a testing dataset, and the remaining 8 subjects serve as the training set. We report the test result of the last checkpoint.

## E.2 PHYSIONET-MI

**Description & Preprocessing.** PhysioNet-MI is a motor imagery dataset, which consists of data from 109 subjects doing trials of 4 different motor imagery tasks. These tasks are motor imagery of the left fist (Class 1), right fist (Class 2), both fists (Class 3), and both feet (Class 4). STELAR applies a low pass filter with a cut-off frequency at 0.3 Hz, sampling rate at 200 Hz, and 4-second window sample (800 data points).

**Evaluation.** As PhysioNet-MI has its own evaluation set, which we regard as the test set. We adopt the proposed cross-validation protocol for validation sets by splitting all subjects into 5 folds. We then conduct 5 fine-tunings, each involving one fold of subjects as a validation set, and the remaining subjects serve as the training set.

## E.3 KAGGLEERN

**Description & Preprocessing.** KaggleERN is an error-related potential dataset, which requires each subject to see letters and numbers (showing 36 possible items on a matrix). Each item of the character is flashed in a random order. All subjects interact with a computer interface, which produces responses to the subject's attention over words. EEG is recorded when subjects observed whether the system correctly or incorrectly responds. This dataset consists of 2 labels: Correct feedback or Erroneous feedback. STELAR applies sampling rate of 200 Hz, and 2-second window sample (400 data points).

**Evaluation.** As KaggleERN has its own evaluation set, which we regard as the test set. We adopt the proposed cross-validation protocol for validation sets by splitting all subjects into 5 folds. We then conduct 5 fine-tunings, each involving one fold of subjects as a validation set, and the remaining subjects serve as the training set.

## E.4 TUEV

**Description & Preprocessing.** TUEV is a seizure detection dataset, which is a subset of TUEG. This dataset records clinical EEG segments of 6 classes: spike and sharp wave (SPSW), generalized periodic epileptiform discharges (GPED), periodic lateralized epileptiform discharges (PLED), eye movement (EYEM), artifact (ARTF), and background (BCKG). STELAR applies a band-pass filter from 0.1 Hz to 75 Hz and a notch filter at 60Hz, sampling rate of 200 Hz, and 5-second window sample (1000 data points).

**Evaluation.** As TUEV has its own evaluation set, which we regard as the test set. We adopt the proposed cross-validation protocol for validation sets by splitting all subjects into 4 folds. We then conduct 4 fine-tunings, each involving one fold of subjects as a validation set, and the remaining subjects serve as the training set.

## E.5 SLEEP-EDFX

**Description & Preprocessing.** Sleep-EDFx is a sleep stage classification dataset, consisting of data from 78 healthy subjects. This dataset contains 5 classes, corresponding to 5 stages of sleep: W, N1, N2, N3, REM. STELAR applies a low-pass filter with a cut-off frequency at 30 Hz, sampling rate: 200 Hz, and 30-second window sample (6000 data points) to Sleep-EDFx.

**Evaluation.** We adopt the proposed subject-wise cross-validation protocol. We split the total dataset into 5 folds with the same number of subjects. We perform 5 fine-tunings, each involving a different fold as a testing dataset, and the remaining 4 folds serve as the training and validation sets. We randomly select training and validation data from these 4 folds, with a val-train ratio of 1:9.

## E.6 TUAB

**Description & Preprocessing.** TUAB consists of 409,455 10-second samples of subjects annotated as normal or abnormal (2-label classification). STELAR applies a band-pass filter from 0.1 to 75 Hz, a notch filter at 50 Hz, sampling rate: 200 Hz, and 10-second window sample (2000 data points).

**Evaluation.** As TUAB has its own evaluation set, which we consider as the test set. We adopt the proposed cross-validation protocol for validation sets. We split all subjects into 4 folds of subjects. We then conduct 4 fine-tunings, each involving one fold of subjects as a validation set, and the remaining subjects serve as the training set. Generally, the train-valid-test ratio is 6:2:2.

# F METRICS DESCRIPTION

In this section, we will provide the details about all metrics we used for evaluating the model's performance.

- **Balanced Accuracy.** Balance Accuracy is usually used to measure the performance of imbalanced datasets. It is defined as the mean of recall of each class in the dataset.
- **Cohen's kappa.** Cohen's kappa is a statistical metric used to measure the level of agreement between two classifiers during classification tasks. In the experiments, one classifier is the true label of the sample.
- **Weighted F1.** Weighted F1 is the average value of the F1-score of all classes, where each class's score is weighted by its number of true instances.
- **AUROC.** AUROC stands for Area Under the Receiver Operating Characteristic curve. AUROC measures the ability of a classifier to distinguish between positive and negative classes, which is often used for binary classification.
- **AUC-PR.** AUC-PR stands for Area Under the Precision-Recall Curve. AUC-PR measures the trade-off between precision and recall across different thresholds.

# G LARGE LANGUAGE MODELS (LLMS) USAGE

We utilized a large language model as a support tool in preparing this paper. Its role was limited to: (i) aiding and polishing the clarity and flow of writing (e.g., rephrasing, improving readability), and (ii) assisting with retrieval and discovery, such as identifying relevant prior work and commonly used methods.

All scientific design choices, methodological decisions, implementation, data analysis, and interpretation of results were made solely by the authors. The LLM did not contribute novel ideas or conduct experiments; it was used only as an assistant for writing and literature awareness.

