# OpenReview forum: "STELAR: Dual-space training EEG Foundation Models for Transferable Representations"
_ICLR.cc/2026/Conference — ICLR 2026 Conference Withdrawn Submission_

### Official Review · Reviewer_aWyF · 2025-10-30

**Soundness:** 2
**Presentation:** 3
**Contribution:** 1
**Rating:** 2
**Confidence:** 5

**Summary:**

The paper presents STELAR, an EEG foundation model designed to overcome the challenges of low signal-to-noise ratio, inter-subject variability, and heterogeneous electrode configurations inherent in EEG data. The authors propose an encoder-centric design that reduces the reliance on large auxiliary components, focusing instead on efficient representation learning through a dual-space pretraining strategy. This strategy includes visible-token alignment, masked-token alignment, and signal-space reconstruction, enabling faster convergence and more effective learning of transferable EEG representations. STELAR employs a spatio-temporal cross-attention encoder to capture both spatial dependencies across EEG channels and temporal dynamics over time.

**Strengths:**

The strengths of this paper lie in its thorough literature review, which provides a comprehensive overview of related works in the field. The figures are well-designed, contributing to the clarity of the concepts presented. Additionally, the writing is clear and easy to follow, making its idea more accessible to readers.

**Weaknesses:**

The main weaknesses of this paper lie in its lack of innovation.

1.The model architecture closely follows the CBraMod framework, utilizing the spatio-temporal Criss-Cross attention structure, and the pretraining tasks are largely borrowed from EEGPT. Although the paper introduces the concept of "dual-space alignment," the pretraining tasks are essentially identical to those of EEGPT, and the application of momentum encoder is also the same. The only notable difference is the differentiation between visible and masked token alignment losses, but this distinction does not significantly enhance the novelty of the approach.

2.Indeed, the idea of making the encoder heavier while keeping the decoder lighter is a well-established concept in the MAE framework, which has been widely adopted in various domains. This approach emphasizes the encoder's role in learning robust, high-quality representations while simplifying the decoder to maintain efficiency and reduce unnecessary complexity. The choice of this architecture is not particularly novel, as it has been effectively utilized in MAE and other similar models.

3.Furthermore, the experiments are relatively simple and fail to thoroughly explore the specific characteristics of the proposed method.

Overall, the work lacks depth and does not meet the innovative and rigorous standards expected at ICLR.

**Questions:**

1. The introduction section of the paper lacks clarity and doesn't provide a strong foundation for the proposed work. Many concepts are referenced, but rather than being clearly articulated within the introduction itself, the authors suggest referring to the appendix for further details. This approach detracts from the flow of the introduction and makes it harder for readers to fully grasp the context and motivation behind the work. For example, the discussion around "large auxiliary modules" of current works is mentioned, but the specific components that make up these modules in the corresponding works are not explicitly identified or explained. Additionally, the reasons why these modules may be problematic are not clearly conveyed. It would be beneficial for the authors to address these points directly within the introduction, ensuring a more comprehensive and self-contained explanation of the paper's contributions and the problems it aims to solve. This would help establish a clearer rationale for the proposed model from the outset.

2. The authors mention linear masked reconstruction, which preserves fidelity to raw signals. This needs further clarification, as the rationale behind its use should be explicitly stated to demonstrate its validity and effectiveness.

3. The authors should compare the performance of the proposed lightweight predictor and reconstructor with the corresponding modules used in other models. Adding this ablation experiment would help demonstrate that the lightweight auxiliary components indeed contribute to improving the model's representation learning.

4. The authors should include an ablation experiment on the model architecture to validate the effectiveness of the spatio-temporal transformer in this work. Just because it is a contribution from previous works, it should not be assumed to have a positive impact in this context without direct verification. Proper validation is needed to confirm its relevance and contribution to the proposed model's performance.

5. The authors should compare the designed masked alignment loss + visible alignment loss with the original alignment loss used in EEGPT. This comparison would help demonstrate the effectiveness of the proposed improvements and highlight their impact on the model's performance.

6. It would be beneficial to see the fine-tuning results after pretraining, as this is a critical performance experiment for EEG foundation models.

7. The experiments on the weight parameters of the three loss functions should be included. This would help observe which loss function plays a crucial role in the model's performance and provide a clearer understanding of the contribution of each loss term to the overall learning process.

8. The comparison models are limited, with only three included. It is recommended to add more models to the comparison, such as BIOT, EEG2Rep, and other deep learning models that are not foundation models. Additionally, baselines in EEGPT, LaBraM, and CBraMod can be included for a more comprehensive evaluation.

---

### Official Review · Reviewer_JKcj · 2025-11-01

**Soundness:** 2
**Presentation:** 2
**Contribution:** 2
**Rating:** 4
**Confidence:** 5

**Summary:**

This paper introduces STELAR, an encoder-centric EEG foundation model that addresses the limitation of existing models dominated by auxiliary components. STELAR uses a dual-space pre-training strategy with three objectives: visible-token alignment, masked-token alignment, and linear masked reconstruction. Built on a spatio-temporal encoder with Criss-Cross attention, STELAR outperforms state-of-the-art models across six EEG tasks, achieving faster convergence (within 15 epochs) and requiring fewer parameters.

**Strengths:**

STELAR's novel approach of focusing on an encoder-centric design is well-motivated and addresses key limitations in current models. The strong empirical results demonstrate significant improvements across diverse EEG tasks, while its efficiency in terms of fewer parameters and rapid convergence makes it practical for resource-constrained environments.

**Weaknesses:**

While STELAR shows solid performance, it lacks visualizations of EEG pattern features, which would help better understand the encoder's learning process. The comparison with smaller classification networks is limited, leaving the relative advantages of STELAR unclear. Key ablation studies, such as the effect of different positional embeddings, Criss-Cross attention, or RoPE in the encoder, are not explored. Additionally, the novelty of the encoder-centric approach could be questioned given prior work in the field, and the overall contribution may not seem sufficiently groundbreaking.

**Questions:**

1. Can you provide more detailed visualizations of the EEG features learned by STELAR? How do these visualizations demonstrate the effectiveness of the encoder in capturing temporal and spatial patterns?
2. Have you compared STELAR against other lightweight classification networks(classification  networks specifically  for each dataset), to better understand its performance relative to simpler architectures?
3. What happens if different positional embeddings are tested in the tiny predictor? How does Criss-Cross attention or RoPE affect performance compared to a standard attention mechanism?
4. How does STELAR's encoder-centric approach compare to other similar models in the literature? Are there aspects of your design that uniquely address limitations in existing models?

---

### Official Review · Reviewer_TWAr · 2025-11-01

**Soundness:** 3
**Presentation:** 3
**Contribution:** 2
**Rating:** 4
**Confidence:** 4

**Summary:**

This paper presents an EEG foundation model pre-trained using a mask-and-reconstruct approach. The model employs criss-cross attention to capture both temporal and spatial dependencies across EEG patches. And it extends the EEGPT framework by incorporating an alignment-based loss into the training objective. Under a linear-probe evaluation protocol, the proposed model demonstrates superior performance compared to other EEG pre-training baselines.

**Strengths:**

The authors provide meticulous documentation of their data-handling pipeline, including well-justified parameter choices and a sound data-cleaning procedure, which enhances the reproducibility of the study.

The empirical results demonstrate that the proposed model achieves competitive performance, outperforming existing baselines in a linear-probe evaluation setting.

**Weaknesses:**

The paper has limited conceptual novelty, as the core architectural and pre-training strategies are adaptations of existing models.

The empirical evaluation is relatively narrow. By restricting the assessment to a linear-probe protocol, the paper fails to demonstrate the model's performance under more practical fine-tuning scenarios, which limits the understanding of its generalizability.

While the data-cleaning process is well-described, the absence of an ablation study quantifying its impact is a notable omission, leaving the necessity and effect of this step unclear.

**Questions:**

Please refer to the weakness

---

### Official Review · Reviewer_yLbm · 2025-11-01

**Soundness:** 2
**Presentation:** 2
**Contribution:** 2
**Rating:** 2
**Confidence:** 4

**Summary:**

The paper introduces STELAR, an encoder-centric EEG foundation model that aims to learn transferable representations through a dual-space pretraining strategy combining representation-space alignment and lightweight signal-space reconstruction. The authors emphasize reducing the complexity of auxiliary components while strengthening the encoder, and report competitive results on multiple EEG benchmarks under a linear probing evaluation.

**Strengths:**

The paper is well-structured and clearly written, with a coherent narrative that effectively communicates the motivation and design of STELAR.

**Weaknesses:**

1. Limited Conceptual Novelty: While the paper positions STELAR as an encoder-centric model with a dual-space objective, the core components—such as the Criss-Cross attention mechanism and the use of a momentum encoder—are borrowed directly from prior work (e.g., CBraMod, EEGPT). The proposed “dual-space alignment” does not represent a substantial departure from existing frameworks, and the incremental nature of the contributions may not meet the bar for ICLR.

2. Insufficient Justification for Design Choices: The paper lacks a thorough ablation study to validate key architectural decisions. For instance: The benefits of the lightweight predictor and reconstructor are not quantitatively compared against heavier variants used in other models. The effectiveness of the spatio-temporal encoder in the STELAR is not experimentally verified. The distinction between visible and masked token alignment is not sufficiently motivated or compared to the original alignment loss in EEGPT.

3. The comparison with baseline models is restricted to only three recent works, omitting other relevant models, which limits the breadth of the evaluation.

**Questions:**

1. Could the authors provide a more detailed motivation for the “dual-space alignment” concept? How does it fundamentally differ from the alignment strategies used in current work such as EEGPT or other models?

2. Please include a comparison between the proposed lightweight auxiliary modules and their heavier counterparts in terms of both performance and parameter efficiency. This would help substantiate the claim that the encoder is the primary contributor to representation quality.

3. An ablation study on the loss weighting (λ_v, λ_m, λ_r) would help clarify the relative importance of each objective and guide future work in balancing alignment and reconstruction.

4. Consider expanding the set of baseline models to include non-foundation model approaches and other recent self-supervised EEG methods (e.g., BIOT, EEG2Rep) to provide a more comprehensive performance context.

---

### Note · Authors · 2025-11-28

I have read and agree with the venue's withdrawal policy on behalf of myself and my co-authors.